# WHAT TO DO IF SPARSE PHYSICAL EQUATION LEARNING FAILS UNEXPECTEDLY?

## ABSTRACT

Learning physical equations from data is essential for scientific discovery and engineering modeling. However, most of the existing methods rely on two rules: (1) learn a sparse representation to fit data and (2) check if the loss objective function satisfies error thresholds. This paper illustrates that such conditions are far from sufficient. Specifically, we show that sparse non-physical approximations exist with excellent fitting accuracy, but fail to adequately model the situation. To fundamentally resolve the data-fitting problem, we propose a physical neural network (PNN) utilizing "Range, Inertia, Symmetry, and Extrapolation" (RISE) constraints. RISE is based on a complete analysis for the generalizability of data properties for physical systems. The first three techniques focus on the definition of physics in space and time. The last technique of extrapolation is novel based on active learning without an inquiry, using cross-model validation. We validate the proposed PNN-RISE method via a synthetic dataset, power system dataset, and mass-damper system dataset. Numerical results show the universal capability of the PNN-RISE approach to quickly identify the hidden physical models without local optima, opening the door for the fast and highly accurate discovery of the physical laws or systems with external loads.

## 1 INTRODUCTION

Internet of Everything (IoE) connects new edge devices into an intelligent web at an unprecedented speed. If utilized systematically, IoE can create much more efficient productivity via coordinated effort among humans, processes, data, and things (Li et al., 2020). Among various analytical frameworks, dynamic data-driven applications systems (DDDAS) naturally interweaves diversified information via data-driven analysis to assimilate the system model for a wide range of applications (Blasch et al., 2018), thus significantly increasing the performance of both the simulation and application models. The simulation model usually incorporates physical methods (Nasiakou et al., 2018; Chowdhury & Subramani, 2020) which provides a structure for model-based and data-driven coordination. However, many of the physical laws in the web of IoE can be unknown, which remain to be discovered manually or automatically to enable down-streaming applications with accurate modeling.

The most economically efficient way of physical law discovery is to use machine learning to analyze the pattern behind the data stream, which is an important topic (Sahoo et al., 2018; Udrescu & Tegmark, 2020). In previous studies, such as symbolic regression (Petersen, 2019), the sparsity representation is used to measure the accountability of physics (Brunton et al., 2016). The underlying assumption is that sparsity can ensure physical exactness due to parsimonious principle (Li, 2013). While low complexity of learning models is a necessary condition for enhancing the understanding of physical laws (Schölkopf et al., 2002; Blasch et al., 2021), there is not a universal sufficient condition to solve model recovery problem with a guarantee. Therefore, one approach is to add additional conditions to ensure a physical recovery, e.g., near-zero fitting errors (Champion et al., 2019) or/and cross validation (Kim et al., 2020). However, it can be observed that different models easily fit the available data with a sparse but wrong model, which can happen even with cross-validation.

The overfitting should challenge the directions of many researchers in physical system discovery, who believe that only the true physical laws can achieve sparse recovery with a perfect fit. To resolve the problem, we first eliminate non-physical local optima of the neural network (NN) by

discarding all the non-physical parameters, i.e., model parameters indirectly related to the physical representation. The process is novel as existing NN-based methods require certain non-physical parameters to control the representation sparsity (Liu & Laili, 2018; Petersen, 2019). We show how to design a NN with pure physical parameters, entitled a *physical neural network* (PNN). The key of PNN is to enforce activation to physical weights for sparse selection, completely removing the non-physical parameters and local optima.

In training the PNN, there can still be local optimum solutions, the actual knowledge of physical system can enhance model fidelity. To obtain the ground truth, physical system examination can be used to extract common and universal properties. Subsequently, the PNN is restricted to systematically obey the physical system properties. Specifically, constraints based on the physical distance and symmetry are embedded in the training function to quantify the spatial relationship enforced by physical law. Furthermore, as physical systems evolve over time, inertial property for physical systems are extracted and updated. In general, the regularization from physical laws focuses the learning/optimization towards a more meaningful fitness understanding of the given data.

Although the two designs of PNN and physics constraints aim to obtain the physical laws via fitting existing data, the incorporation of physical laws' biggest impact is its capability to achieve zero fitting errors even beyond the training data range. Unfortunately, it will be costly if not impossible to conduct active learning for every Internet of Things (IoT) system due to its scale. To provide the "free lunch" of adding the extrapolation capability of PNN, we propose a new learning method for cross-model validation based on probabilistic estimates with physical kernels. We use a physically meaningful forecast to identify out-of-sample points with a confidence level. Then, the results are integrated as a weak supervision for learning the exact physics for extrapolation, in addition to the constrained PNN. Together, PNN-RISE uses the PNN container to include the RISE principles, enforcing the recovered function to satisfy constraints on *range (R), inertia (I), symmetry (S), and extrapolation (E)*. Empirically, extensive validations show that the PNN-RISE outperforms the state-of-the-art methods. Consistently, scenarios with or without each constraint in RISE validate the approach. Furthermore, for each constraint, we quantify its strength with different levels, highlighting that loose constraints are enough to obtain high accuracy in the proposed PNN.

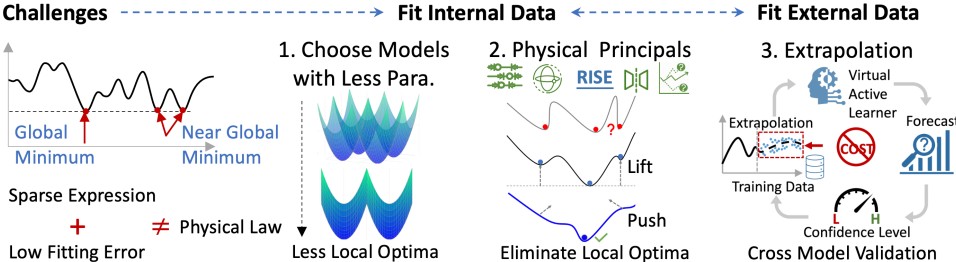

Figure 1: The proposed framework.

In summary, the contributions in Fig. 1 are 1) showing that the sparse recovery with low training and testing errors does not ensure physical law recovery, 2) reducing the possibilities of local optima significantly by eliminating non-physical parameters in learning model selection and design, 3) proposing four physical principles to completely characterize physical systems via constrained learning, and 4) creating a novel learning method for physical systems extrapolation as probabilistic forecasting out-of-sample cross-model validation (FOCV).

## 2 RELATED WORK

**Physical System Identification.** To recover system information, supervised learning (SL) can approximate input-output relationships for inferences (Huang et al., 2006; Hornik et al., 1990). They show good numerical performance in non-physical and physical applications, including computer vision tasks and natural language processing (Lu & Weng, 2007; Wang et al., 2017a; Karpathy et al., 2014; Nadkarni et al., 2011; Hirschberg & Manning, 2015). Deep neural networks (DNNs) boost classification accuracy (Hinton et al., 2006; Krizhevsky et al., 2012) by using latent layers to extract nonlinear features hidden in images. While it makes SL useful for non-physical systems,

physical systems require a much higher accuracy for operation or control, e.g., near 100% for Six Sigma performance (Walshe et al., 2010), so that operators can safely operate the system to avoid outages or cascading failures (Chung et al., 2018; Liu et al., 2017). Even worse, many SL models are black boxes that lack physical interpretability for reliable operations, as well as certifiability for performance standards (Mestav et al., 2018; Du & Li, 2019; Hu et al., 2020; Baghaee et al., 2017).

Therefore, some works find utilizing contextual information improves physical-system identification where they assume a good knowledge of physical basis and sufficient measurements across the system (Yu et al., 2017; Chen et al., 2020). Similarly, (Brunton et al., 2016; Champion et al., 2019; Li & Weng, 2021) assume prior symbols are known in a dictionary and introduce sparse regression to select symbols. However, the assumptions are too limiting due to the difficulty of knowing all the physical basis for arbitrary complex systems (Wang et al., 2017b; Tarca et al., 2007). One ad-hoc approach (Schmidt & Lipson, 2009; Petersen, 2019) uses a genetic algorithm to search for the symbolic basis but the problem is NP-hard without any guarantee on recovering the governing physical laws from empirical data. Thus, we investigate whether the sparsity technique and a small modeling error are sufficient to claim the success of learning the physical model. Moreover, how can analytical techniques ensure exact physics learning for domain-specific systems?

**Constrained Machine Learning.** There are many applications of constrained machine learning (CML) in different systems. (Small et al., 2011) restricts the weight vectors with domain knowledge in support vector machine and (Li et al., 2021) proposes constrained adversarial learning to generate adversarial examples under physical constraints. In deep learning applications, (Zhao et al., 2019) employs constraints from conservation law to restrict the output variables of DNN. (Velloso & Van Hentenryck, 2020; Fioretto et al., 2020a) solve optimal power flow problems using DNN with physical constraints, where the output variables are biased to guarantee feasible solution. (Tran et al., 2020) restricts the input to DNN into different groups to guarantee privacy. Among these studies, (Fioretto et al., 2020b) optimizes DNN with output constraints using Lagrangian duality. These CML methods can solve their own problems effectively; however, there is a need for general principles to embed the physical constraints systematically with efficient solutions.

**Data Forecasting.** For previous data forecasting methods, autoregressive moving average (ARMA) and autoregressive integrated moving average (ARIMA) predict the evolving variables based on their lagged values (Gilbert, 2005). To capture nonlinear correlations in the short-term and long-term, there are many deep learning-based methods, e.g., recurrent Neural Network (RNN), long short-term memory (LSTM), bidirectional LSTM (BiLSTM), gated recurrent units (GRUs), and variational autoencoder (VAE) (Zeroual et al., 2020). To forecast the result with confidence, methods like Gaussian process (GP) (Quinonero-Candela & Rasmussen, 2005), Quantile Regression (Maciejowska et al., 2016) and deep learning-based models (Salinas et al., 2020) afford an efficient way to predict distributions for the future data. Since the uncertainty evaluation is important to integrate the forecast data into the physical learning process, probabilistic methods are presented to estimate probability density functions in the proposed FOCV.

## 3 METHODS

The PNN-RISE approach is to understand the reason why a "good" sparse fitting is not physical verifiable, to design physical principles to guide the fitting internally based on a flexible PNN structure, and to create a cost-free active learning method for extrapolation at unseen operating points.

Symbolic regression (SR) is a common approach to find equation operators ($\{+, -, \times, /\}$) and expressions like ($x^2, \cos(x)$), etc. Existing papers (Brunton et al., 2016; Sahoo et al., 2018; Champion et al., 2019; Petersen, 2019; Kim et al., 2020) show that if the following three conditions are satisfied, SR is capable to find the true equation: 1) apply the sparse regularization, 2) achieve nearly zero training error, and 3) utilize cross validation to avoid overfitting. However, we observe that SR methods are prone to non-physical solutions even when all three assumptions are satisfied. For example, to learn the function $y = x_1 x_2 + x_2 x_3$ from data, different SR methods can easily get trapped in local optima. To see if these observations are special, we test different cases, where Fig. 2 shows a table summary of four different cases, with integer coefficients or not. We also test different numbers for multiplication and summation. From all examples, one can see that the recovered objective has sparse coefficients and negligible error (loss objective $\mathcal{L}$) at non-desirable local optimums. Other than the toy examples, we conduct a realistic power system case study and visualize the objective

surface in Appendix A. Similarly, for a small parameter range, there are points achieving the global optimum or local optima that are very close to the global optimum. However, only one of them is the physical ground truth. For example, one can go from the red point to a non-physical solution with an error close to zero, even when the sparsity regularization is enforced. Therefore, we can see that "good" sparse fitting is insufficient to avoid overfitting. Thus, we need to avoid overfitting for physical systems not only by **choosing sparse learning models**, but also new ways to constrain the learning process and avoid non-physical solutions.

| Solutions | Case 1 | Case 2 | Case 3 | Case 4 |
|---|---|---|---|---|
| Physical Truth | $y = x_1x_2 + x_2x_3$ 
 $\mathcal{L} = 0.00$ | $y = 0.1x_1x_2 + 0.3x_2x_3$ 
 $\mathcal{L} = 0.00$ | $y = x_1x_2x_3 + x_2{}^2x_3$ 
 $\mathcal{L} = 0.00$ | $y = x_1x_2 + x_2x_3 + x_1x_3$ 
 $\mathcal{L} = 0.00$ |
| Local Optimum 1 | $y = 0.0024x_1x_2 +$ 
 $2.0793x_1^{0.4945}x_2x_3^{0.4723}$ 
 $\mathcal{L} = 0.002$ | $y = -1.3192x_1^{0.6051}x_2^{0.3718}x_3^{0.3445}$ 
 $+1.1166x_1^{0.3569}x_2^{0.8011}x_3^{0.6605}$ 
 $\mathcal{L} = 0.003$ | $y = 6.2041x_1^{0.4331}x_2x_3^{0.9930}$ 
 $8.2098x_1^{0.4833}x_2x_3^{0.9993}$ 
 $\mathcal{L} = 0.021$ | $y = 1.8595x_1^{0.7805}x_2^{0.6445}x_3^{0.6502} +$ 
 $0.3481x_1^{0.4670}x_2^{-0.5860}x_3^{0.3862}$ 
 $+2.2448x_1^{0.1563}x_2^{0.7454}x_3^{0.7202}$ 
 $\mathcal{L} = 0.002$ |
| Local Optimum 2 | $y = 1.5205x_1^{0.4126}x_2^{0.9995}x_3^{0.3445}$ 
 $+0.9600x_1^{0.5369}x_2^{0.9920}x_3^{0.3693}$ 
 $\mathcal{L} = 0.001$ | $y = -2.6189x_1^{0.1352}x_2^{0.6277}x_3^{0.5121}$ 
 $+1.4484x_1^{0.2075}x_2^{0.8744}x_3^{0.6692}$ 
 $\mathcal{L} = 0.003$ | $y = 10.4416x_1^{0.7975}x_2x_3^{0.9997}$ 
 $-9.8486x_1x_2^{0.6719}x_3^{0.9996}$ 
 $\mathcal{L} = 0.003$ | $y = 1.8902x_1^{0.6901}x_2^{0.4758}x_3^{0.7413}$ 
 $1.0969x_1^{0.6724}x_2^{0.8458}x_3^{0.5806}$ 
 $1.0300x_1^{0.1140}x_2^{0.2962}x_3^{0.9306}$ 
 $\mathcal{L} = 0.002$ |
| Near Global Optimum | $y = x_1x_2 +$ 
 $x_2^{0.9900}x_3^{0.9900}$ 
 $\mathcal{L} = 0.026$ | $y = 0.0990x_1x_2 +$ 
 $0.3100x_2^{0.9900}x_3^{0.9900}$ 
 $\mathcal{L} = 0.004$ | $y = x_2^{1.9800}x_3 +$ 
 $x_1^{0.9900}x_2^{0.9900}x_3^{0.9900}$ 
 $\mathcal{L} = 0.056$ | $y = x_1x_2 + x_2x_3 +$ 
 $x_1^{0.9900}x_3^{0.9900}$ 
 $\mathcal{L} = 0.018$ |

Figure 2: Absolute mismatches between the recovered functions and the physical ground truth.

### 3.1 Proposed method: physical neural network with weight activation

For complex physical systems, the governing functions are formed by basic operations such as additive and multiplicative accumulations of variables. For example, the topology of power system can be represented as a graph, where the interaction between different nodes along the lines is represented by polynomials of the voltages for power flow (Yu et al., 2017). To combine the benefit of universal approximation and the freedom to choose activation function, we propose to use a NN as a container to learn the representation of physical equations, and thus call it a physical neural network (PNN), denoted as $f$. To train $f$, the training data set is $\{\boldsymbol{x}_i, \boldsymbol{y}_i\}_{i=1}^N$, where $\boldsymbol{x}_i$ and $\boldsymbol{y}_i$ are the $i^{th}$ input and output of $f$, and $N$ is the number of training samples.

To construct the physical equation in PNN, the key is to develop a sparse selection mechanism to automatically select part of the variables to join summation/subtraction and multiplication/division. For sparse selection, existing work utilizes sparse regularization like LASSO for weight updates (Kim et al., 2020), introduces extra neural network modules like gates (Liu & Laili, 2018), or conducts symbolic generators (Petersen, 2019). In general, they demand the quantification of extra hyper-parameters (i.e., the LASSO penalty) or parameters (i.e., parameters in gates or symbolic generators), which significantly increases the possibility of local optima.

In order to address the problem towards limiting local optima, we advocate the design philosophy: enforce activation functions on weights for the sparse selection of features. Thus, no extra parameters are added, and the searching is only for physical parameters. To elaborate on our design, we need to answer (1) which parameters need to be activated and (2) how to enforce the activation. For parameter activation, we start from the basic symbolic operations in neural networks, e.g., multiplications and divisions of the form $z_j = \Pi_i^N x_i{}^{p_{ij}}$, where $z_j$ is the $j^{th}$ element of the layer next to the input. $x_i$ is the $i^{th}$ element of the input vector variable $\boldsymbol{x}$ and $p_{ij}$ is the polynomial order number (power) of $x_i$. Note $p_{ij} \in \{-K, \cdots, -1, 0, 1, \cdots, K\}$, where the positive integer represents the multiplication, the possibility of $0$ corresponds to the sparsity, and the negative integer represents the division. Then, each $z_j$ is a more complex symbol for the next operation and $p_{ij}$ is the optimization variable. To accommodate $p_{ij}$ into a NN, we adopt logarithm and exponential operators to achieve a linear summation format $z_j = \exp\left(\sum_{i=1}^N p_{ij} \ln(x_i)\right)$ (Trask et al., 2018). Thus, the linear layer with weights $p_{ij}$ needs to be activated for arithmetic operations and sparsity. Considering the target integer range of $p_{ij}$, it's vital to know how to control the values for parameter activation.

We first show how to enforce $p_{ij} \in \{-1, 0, 1\}$. Since tanh function can restrict the output value to be in $[-1, 1]$, we can utilize a positive scaling factor $c_1$ and tanh to activate some free and optimizable parameter $w_{ij}$ such that $tanh(c_1w_{ij}) \in \{-1, 1\}$, where the scaling is to make sure the output is very close to the target $-1$ or $1$. Subsequently, the network needs to enforce $0$ for sparsity. We

similarly create another sigmoid function to achieve $\sigma(c_2 w_{ij}') \in \{0, 1\}$, where $\sigma$ is the sigmoid function, $c_2$ is the positive scaling factor, and $w_{ij}'$ is another optimizable parameter. To summarize, we have $p_{ij} = \tanh(c_1 w_{ij}) \cdot \sigma(c_2 w_{ij}')$, where $p_{ij} \in (-1, 1)$. The multiplication of tanh and sigmoid functions not only generates a smooth optimization surface but also naturally results in limited parameters. After that, a third linear layer is used to obtain $y_k = \sum_j m_{jk} z_j$, where $m_{jk}$ denote the layer weights for linear summation. Since the summation and abstraction are incorporated into the linear layer, we completely represent $\boldsymbol{y} = f(\boldsymbol{x})$. Using such a design as a basic unit, we can extend from $\{-1, 0, 1\}$ to $\{-K, \cdots, -1, 0, 1, \cdots, K\}$ for different physical systems.

Such a design converts the non-convex multiplication to a convex form of linear summation for NN training. Based on this design, we show the following property of our PNN to guarantee the finding of stationary points in the optimization. The following proposition states the finding of stationary points using PNN with the design and the proof is included in Appendix.

**Proposition 1** (Biconvexity of PNN). *PNN is biconvex on the parameters of the logarithm and the linear summation layers. The stationary points for physical function can be found in optimization.*

Proposition 1 indicates that PNN can find the stationary points (i.e., saddle points, local optima, and global optima). Subsequently, can we find the global optima and can it represent the true physics?

## 3.2 Embed physical principles via constraining PNN

After providing a flexible design to limit possibilities of local optima, we need comprehensive principles to avoid non-physical local optima. If the derived constraints can be convex, they will provide guarantees in solving or reducing the issue of non-physical local optima.

For physical systems, two important perspectives are physical existence and systematic behaviors. As the components are physically located, there are geographical or electrical distances, etc. Therefore, one common constraint through physical systems is the symmetric (S) property, e.g., a branch connecting two components represents a common coupling. In addition, the links between different components also have their physical limitations, leading to a range (R) constraint. At the system level, the physical system will continue making an impact to the environment, leading to an inertia (I) concept, which is the momentum from a physical object because physical systems typically have natural inertia, whether it is strong or weak. For man-made systems, there could be additional artificial inertia due to factors such as guaranteeing stability for sustainable operation, e.g., in power systems. Finally, such inertia does not only live in the temporal space but also in the input/output data space. For example, the physical equations are typically continuous functions. These concepts open the door to investigate the data range beyond the training and testing data. Such generalized inertia leads to an important extrapolation (E) property across different physical systems to forecast information yet to be observed but representative of plausible scenarios. The constraints based on different properties of physical systems are called RISE concept by combining their initials, which is demonstrated via a figure in Appendix. This subsection introduces the first three to constrain PNN in Subsection 3.1 while Subsection 3.3 introduces the FOCV method for physical system extrapolation.

**Range (R): Shape Constraint on Physical Parameters:** While local optima generate many possible parameters to minimize the prediction error, the true physical parameters of corresponding features usually yield specific shapes (Cotter et al., 2019). For example, the range of power line admittance (reciprocal of impedance) of power distribution systems can be specified with voltage level and line materials. In addition to the bound on value range of independent parameters, there can also be shape constraints on the conditional relationships among parameters, e.g., longer lines have larger resistance than shorter lines. Such a rule specified by the physical shape constraint function can filter out many local optima even with low errors. In order to enforce the learned physical parameters $W_p$ within the constraint, a customized gate function is proposed to filter the weights: for $g(W_p) \leq \lfloor g_p \rfloor$, $g'(W_p) = \lfloor g_p \rfloor$; for $\lfloor g_p \rfloor \leq g(W_p) \leq \lceil g_p \rceil$, $g'(W_p) = g(W_p)$; and $g'(W_p) = \lceil g_p \rceil$ for $g(W_p) \geq \lceil g_p \rceil$. $g(\cdot)$ is a function of physical weights $W_p$ to represent the type of shape constraint. As an example, $g(\cdot)$ is linear for specific value constraint of independent parameters, where $\lfloor g_p \rfloor$ and $\lceil g_p \rceil$ are the lower and upper bounds, and $g'(W_p)$ are the constrained weights.

**Symmetry (S): Constraints on Physical Couplings:** For physical systems with multiple outputs, there are variable couplings among different outputs, e.g., one term is contained in two outputs. Let's use the power system as an example. A certain amount of power flowing from one place to

another means that the same amount of power will be subtracted and added simultaneously at the two places. Based on our observations, most of the local optima of NN do not satisfy such a rule. Thus, the property is embedded by minimizing $L_{Mutual} = \|W_p - W_p^T\|_2 + \|\text{rank}(W_{p_i}) - N_{mutual}\|_1$. The first term is an $l^2$ norm penalty on the symmetry of recovered physical parameters and the second term is to further ensure that the number of mutual terms $N_{mutual}$ satisfies the system property, e.g., power system equation has $N_{mutual} = 2$, because two nodes are connected by one power line. For practical implementation, the $l^0$ norm is relaxed to $l^1$ norm in optimization.

**Inertia (I): Constraints on Physical Tendency:** For physical systems, the states are typically continuous and cannot change suddenly, leading to physical inertia. Such a property can be embedded by bounding the outputs of two adjacent time slots. To ensure the partial convexity, the Euclidean norm is modeled as a penalty in loss function $L_{Inertia} = \|f(\boldsymbol{x}_{t+1}) - f(\boldsymbol{x}_t)\|_2$.

### 3.3 CROSS-MODEL VALIDATION FOR PHYSICAL EXACTNESS AT UNKNOWN DATA REGION

Subsections 3.1 and 3.2 show how to avoid overfitting by considering the NN-based model design and fitness of data to the physical laws. However, all these designs must be within the training data range. *What will happen beyond the training data range, e.g., a new operation point of physical systems unseen in the past?* Actually, determining the performance region is the most useful part for physical systems modeling after understanding the physical laws, e.g., for universal data fitting at arbitrary points. For example, Fig. 3 shows three globally optimal or near globally optimal lines (red, blue, and green) that can fit data equally well within the training dataset. One way to avoid the selection dilemma is to conduct an active inquiry out of the historical data range for a selection. For example, Fig. 3 illustrates that a few out-of-sample data points in grey can regularize the learning to the ground truth (dotted black line). However, the costs of such an inquiry can be quite expensive as one inquiry may not be enough. Also, given the numerous networks that the internet of everything (IoE) has, it is infeasible and sometimes impossible to conduct an inquiry for out-of-sample estimation. Therefore, *how can a learning approach* avoid any cost of active learning but still explore the generalization capability via some alternative active learning?

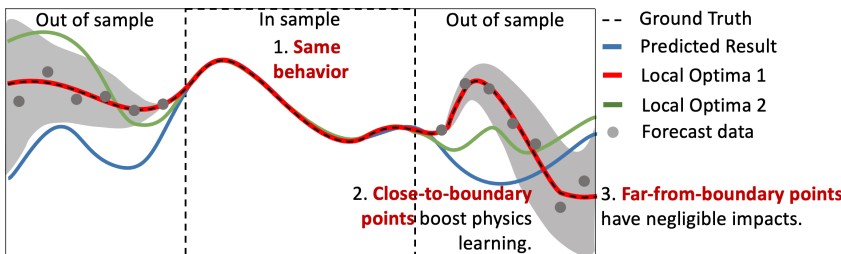

Figure 3: Illustration of out-of-sample forecasting to improve extrapolation.

**Extrapolation (E): Constraints on Physical Validation at Unknown Regions:** To conduct active learning without an inquiry cost, we implement physically meaningful forecasts via a probabilistic model for data points not existing in historical measurements. The learning objective is to provide another level of physical guidance for the constrained PNN in previous subsections. Specifically, an ensemble of forecasts is conducted for both the mean and variance, as the forecast ability decreases when the inquiry point is far away from the data range. Another important design is to embed the physical form into the forecast so that the model can cross-validate the learning models. We propose to utilize Gaussian Process (GP), which not only has computational efficiency in probabilistic forecasts but also provides the flexibility to embed physical forms into its kernel function.

For training the GP, the data is represented as $\{\boldsymbol{x}_i, \tilde{\boldsymbol{x}}_i\}_{i=1}^N$, where $\boldsymbol{x}_i$ represents the input samples and $\tilde{\boldsymbol{x}}_i$ represents a sample point $k$-step later than $\boldsymbol{x}_i$. Based on such data, GP assumes prior function values in a Bayesian framework, e.g., $p(\boldsymbol{g}|\boldsymbol{x}_1, \cdots, \boldsymbol{x}_N) = \mathcal{N}(\boldsymbol{0}, K)$, where $\boldsymbol{g}$ is a vector of latent function values, $\mathcal{N}$ represents normal distribution, and $K$ is a kernelized covariance matrix. In particular, we can add different kernel mappings to calculate $K$ based on our priors of the physical knowledge. For example, for power systems, the polynomial of order 2 is often utilized to represent the relationship between voltages and powers (Yu et al., 2017). Using data with kernel knowl-

edge, we obtain a predicting function via Bayes rule $p(\boldsymbol{g}^{\star}) = \frac{1}{p(\tilde{\boldsymbol{x}})} \int p(\tilde{\boldsymbol{x}}|\boldsymbol{g})p(\boldsymbol{g}, \boldsymbol{g}^{\star})d\boldsymbol{g}$, which has a closed-form solution $\boldsymbol{g}^{\star}$ based on Gaussianity. The goal is to predict the $k$-step later points for any input sample. Specifically, one can plug-in the input data $\boldsymbol{x}$ and obtain the distribution of the $k$-step points $p(\boldsymbol{g}^{\star}(\boldsymbol{x}))$. Thus, the GP-based forecasting not only brings the forecast point $\tilde{\boldsymbol{x}}_k$, e.g., the mean of $p(\boldsymbol{g}^{\star}(\boldsymbol{x}))$, but also the confidence level $\alpha_k$ based on the standard deviation. Similarly, we can obtain $\tilde{\boldsymbol{y}}_k$ and $\beta_k$, where $\tilde{\boldsymbol{y}}_k$ is the forecast point for the output data $\boldsymbol{y}_k$ and $\beta_k$ is the corresponding confidence level. Fig. 3 illustrates the results of GP, where the forecast points have smaller uncertainty if they are closer to the dataset boundary.

With the forecast results from above, we can add $\tilde{\boldsymbol{x}}_k$ ($\tilde{\boldsymbol{y}}_k$) to the training data set, like active learning. However, our method is different from active learning, which is associated with a cost. With the cost, active learning can provide a highly accurate label. But, our new data points are always associated with confidence bounds. The good news is that the FOCV method can provide infinite data points without a cost. Therefore, different than active learning, FOCV uses all the data points with unequal weights, e.g., the inverse of confidence level, to achieve model selection. Therefore, we propose an integral-based regularization for extrapolation $L_{Extrapol} = \frac{1}{\alpha_k \beta_k} \sum_{k=1}^{K} (\tilde{\boldsymbol{y}}_{\boldsymbol{k}} - f(\tilde{\boldsymbol{x}}_{\boldsymbol{k}}))^2$. With extrapolation (E), the proposed methods complete the derivation of RISE. It's important to note that in RISE, the factors are decoupled out of the PNN. Furthermore, the factors can be relaxed into a convex form, which significantly increases the probability of finding the global optima. Specifically, Theorem 1 highlights the regularization can lead to strong convexity over a set of closed and convex regions of the parameter space, i.e., the piecewise strong convexity.

**Theorem 1** (Piecewise strong convexity of PNN with regularization). *With inertia and symmetry regularization, the loss function of the PNN is piecewise strongly convex in the parameter space.*

*Proof.* For the derivation simplicity, we denote $W$ as the parameter set of PNN and denote $l(W)$ as the loss function, i.e., mean square error. The range constraint (R) and the extrapolation constraint (E) are included in $l(W)$ as they are included in the PNN representation. Subsequently, the inertia (I) and symmetry (S) regularizations help to define $l_1(W) = l(W) + \lambda(L_{Inertia} + L_{Mutual})$ as the regularized loss function for training, where $\lambda$ is a positive penalty term. To determine convexity, we write the second derivative of the regularized loss function $\frac{d^2}{dt^2}|_{t=0}l_1(W + tX) = \frac{d^2}{dt^2}|_{t=0}l(W + tX) + \lambda h(W)$, where $h(W)$ is the second derivative of $L_{Inertia} + L_{Mutual}$. $L_{Inertia}$ has a quadratic form to be convex. For symmetry $L_{Mutual}$, the $l^1$ norm regularization is convex and $h(W) > 0$. However, due to the non-convexity of neural network, the first term $\frac{d^2}{dt^2}|_{t=0}l(W + tX)$ is generally not positive semi-definite. The following lemma in (Milne, 2018) proves a lower bound for $\frac{d^2}{dt^2}|_{t=0}l(W + tX)$ with respect to the loss value of $l$.

**Lemma 1.** *Suppose $||\boldsymbol{y}_i||_2 \leq r, r > 0, \forall\, 1 \leq i \leq N$. The second derivative of $l$ in direction $X$ satisfies $\frac{d^2}{dt^2}|_{t=0}l(W + tX) > -\sqrt{2}H(H-1)||W||_{\star}^{H+1}||X||_2^2 r l(W)^{1/2}$, where $H$ is the number of hidden layers in a neural network, $||\cdot||_{\star}$ is a defined norm such that $||W||_{\star} = \max_{0 \leq i \leq H} ||W_i||_2$.*

According to Lemma 1, the second derivative of $l$ is lower-bounded by a term that depends on $l$ value. Then, for regions with small loss values, the positive value $h(W)$ helps to restrict $\frac{d^2}{dt^2}|_{t=0}l_1(W+tX)$ to a positive number, which determines the shape of the region to be convex. Mathematically, we define the region $U(\lambda, \theta)$, where $\lambda > \theta > 0$, by $U(\lambda, \theta) = \{W | l(W)^{1/2}||W||_{\star}^{H-1} < \frac{(\lambda-\theta)h(W)}{\sqrt{2}H(H+1)r||X||_2^2}\}$. It's obvious that $\forall W \in U(\lambda, \theta), \frac{d^2}{dt^2}|_{t=0}l_1(W + tX) > 0$. Namely, within the region of $U(\lambda, \theta)$, the regularized loss function is strongly convex. In general, $U(\lambda, \theta)$ is not a convex set since many local optima points are disconnected and have small $l$ values. However, based on measure theory, we can utilize a set of closed convex sets to completely cover $U(\lambda, \theta)$, leading to piecewise strong convexity of PNN over these closed sets. Therefore, the piecewise strong convexity from RISE regularization guarantees that we can find the global optimal point for each sub-region, and the global optima satisfies constraints for physical parameters. $\square$

## 4  EXPERIMENTS

**Data Preparation.** In experiments, we first use synthetic datasets for illustration and introduce real physical system datasets of power systems and mass-damper systems as the underlying physical

systems for model training and comparison. The datasets are summarized as follows: 1) **Synthetic dataset** is generated from quadratic functions with adding random noises. 2) **Power system datasets:** The full training datasets of nodal voltage states and power injections are generated from power flow simulations using real feeder models (IEEE 8-bus system and Utility feeder), one-year power consumption profiles, and distributed photovoltaics generation profiles (Yu et al., 2017). 3) **Motion dynamics dataset:** We simulate the dynamic process of the mass-dampers systems with 10 nodes and obtain the measurements of velocity and momentum for the test of physical equation discovery. 4) **Aerodynamic dataset:** The aircraft sensor data for weight estimation is collected from the aircraft simulation based on the aerodynamic model in (Chen et al., 2018).

**Performance Evaluation Metrics.** The following metrics based on (Blasch & Sung, 2021) are used to evaluate the performance of learning physical system governing equations. Prediction error is to evaluate the modeling accuracy and extrapolation capacity of learning model via mean square error (MSE) of the output prediction. Physical exactness (PhyE) is to evaluate how much physics is extracted by comparing the recovered symbolic function and related parameters with true physical equation. Robustness to noise ratio (RNR) is proposed to average all results tested under different signal-to-noise ratios for practical consideration. More details are provided in Appendix.

**Benchmark Methods and Testing Scenarios.** The following benchmark methods are used for comparison: (1) sparse identification of nonlinear dynamics (SINDy) (Brunton et al., 2016), (2) support vector regression (SVR) (Yu et al., 2017), (3) deep residual network (Resnet) (He et al., 2016), and (4) equation learner (EQL$^{\div}$) (Sahoo et al., 2018). Different scenarios are generated for testing: $S_1$ denotes the regular setup of dataset size and noise level (independent Gaussian noise with zero mean and 0.01 standard deviation); $S_2$ denotes a noisy setup with noises of 0.1 standard deviation; $S_3$ denotes a data-limited setup with $10\%$ samples of original dataset available. The results

| $S^*$ | Data | SVR | Resnet | SINDy | EQL$^{\div}$ | PNN-RISE |
|---|---|---|---|---|---|---|
| | Sythetic | $0.003 \pm 0.00$ | $0.005 \pm 0.00$ | $\mathbf{0.001 \pm 0.00}$ | $0.003 \pm 0.00$ | $\mathbf{0.001 \pm 0.00}$ |
| | PS 1 | $0.006 \pm 0.00$ | $0.01 \pm 0.00$ | $0.004 \pm 0.00$ | $0.007 \pm 0.00$ | $\mathbf{0.001 \pm 0.00}$ |
| $S_1$ | PS 2 | $0.05 \pm 0.01$ | $0.02 \pm 0.01$ | $0.03 \pm 0.02$ | $0.07 \pm 0.05$ | $\mathbf{0.01 \pm 0.01}$ |
| | Motion | $0.002 \pm 0.00$ | $0.003 \pm 0.00$ | $0.006 \pm 0.00$ | $\mathbf{0.001 \pm 0.00}$ | $\mathbf{0.001 \pm 0.00}$ |
| | Aero | $\mathbf{0.003 \pm 0.00}$ | $0.005 \pm 0.00$ | $0.008 \pm 0.01$ | $\mathbf{0.003 \pm 0.00}$ | $0.002 \pm 0.00$ |
| | Sythetic | $0.004 \pm 0.00$ | $0.007 \pm 0.00$ | $\mathbf{0.002 \pm 0.00}$ | $0.006 \pm 0.00$ | $\mathbf{0.001 \pm 0.00}$ |
| | PS 1 | $0.005 \pm 0.00$ | $0.01 \pm 0.00$ | $0.007 \pm 0.00$ | $0.009 \pm 0.00$ | $\mathbf{0.001 \pm 0.00}$ |
| $S_2$ | PS 2 | $0.06 \pm 0.03$ | $0.04 \pm 0.02$ | $0.09 \pm 0.04$ | $0.10 \pm 0.02$ | $\mathbf{0.01 \pm 0.01}$ |
| | Motion | $0.003 \pm 0.00$ | $0.005 \pm 0.00$ | $0.009 \pm 0.00$ | $\mathbf{0.001 \pm 0.00}$ | $\mathbf{0.001 \pm 0.00}$ |
| | Aero | $\mathbf{0.003 \pm 0.00}$ | $0.02 \pm 0.01$ | $0.01 \pm 0.00$ | $0.004 \pm 0.00$ | $\mathbf{0.003 \pm 0.00}$ |
| | Sythetic | $0.006 \pm 0.00$ | $0.010 \pm 0.00$ | $0.002 \pm 0.00$ | $0.008 \pm 0.00$ | $\mathbf{0.001 \pm 0.00}$ |
| | PS 1 | $0.008 \pm 0.02$ | $0.02 \pm 0.00$ | $0.004 \pm 0.00$ | $0.01 \pm 0.00$ | $\mathbf{0.003 \pm 0.00}$ |
| $S_3$ | PS 2 | $0.07 \pm 0.03$ | $0.04 \pm 0.01$ | $0.03 \pm 0.02$ | $0.07 \pm 0.05$ | $\mathbf{0.02 \pm 0.01}$ |
| | Motion | $0.004 \pm 0.00$ | $0.007 \pm 0.00$ | $0.006 \pm 0.00$ | $\mathbf{0.001 \pm 0.00}$ | $\mathbf{0.001 \pm 0.00}$ |
| | Aero | $\mathbf{0.003 \pm 0.00}$ | $0.007 \pm 0.00$ | $0.01 \pm 0.00$ | $\mathbf{0.003 \pm 0.00}$ | $0.002 \pm 0.00$ |

Figure 4: Performances of mapping recovery on different datasets under different testing scenarios.

are summarized in Fig. 4. Although other methods can have low errors in some testing cases, they can not maintain the good performance as PNN-RISE does for all the datasets. Especially, EQL$^{\div}$ and SINDy's performance deteriorates when the problem size becomes larger, e.g., power system datasets with data from many nodes. While SVR is robust against noises, it can not recover the nonlinear physical function exactly to further decrease the learning error. In this case, the proposed method shows low learning error and a stable performance in both noisy and data-limited scenarios. Moreover, the performance of function recovery is compared in Fig. 5 (left). The bar plot shows the recovery rates (percentage of runs that correctly recover the ground truth symbolic expression) and the line plot with shades shows the recovery accuracy of the corresponding physical parameters. While EQL$^{\div}$ easily fails in large power system cases, PNN-RISE has high success rate in true function recovery, merely trapped into local optima. For physical parameter recovery, PNN-RISE not only has higher average accuracy but also has lower standard deviation than EQL$^{\div}$.

## 4.1 ABLATION STUDY FOR PHYSICAL CONSTRAINTS

In order to show the performance of using different physical constraints, we provide an ablation study of the proposed NN. We first use the synthetic example to visualize the effect of constraints and show in Appendix. Fig. 5 (right) presents the learning results on power system datasets (IEEE 8-bus system). The performance is evaluated in different test scenarios: 1) regular data acquisition, 2) limited input range (voltages within $[0.98, 1.01]$ for training and $[0.90, 1.10]$ for testing), 3) out-

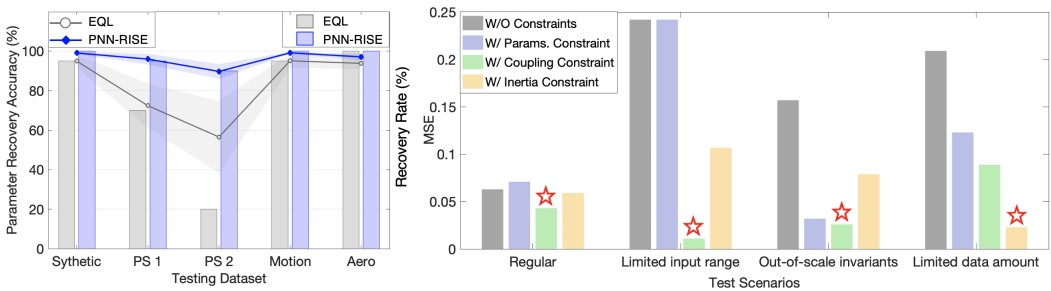

Figure 5: Left: Comparison of physics recovery. Right: Results of embedding different constraints.

of-scale invariants in a governing function, and 4) limited data amount. Generally, adding physical constraints boosts the performance in different testing scenarios. The constraint on mutual variables significantly reduces the learning error by pushing the features and parameters to be symmetric for multiple-output problems. When the data amount is very limited to train NN efficiently, the bound on the inertia of system outputs in time sequence improves the function recovery.

## 4.2 ILLUSTRATION OF THE EFFECTIVENESS OF FORECASTING-BASED PHYSICS LEARNING

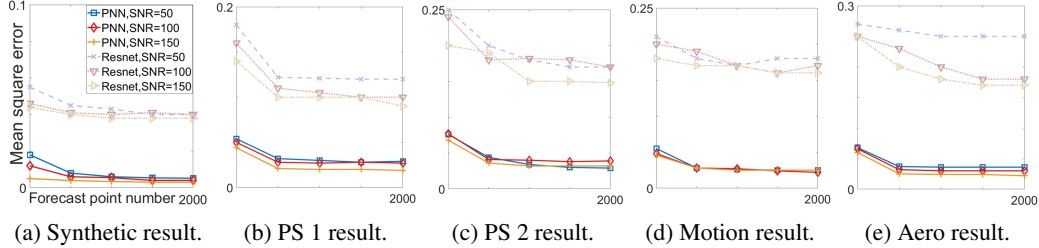

(a) Synthetic result.    (b) PS 1 result.    (c) PS 2 result.    (d) Motion result.    (e) Aero result.

Figure 6: The results of GP-based PNN and GP-based Resnet for different systems.

To illustrate the effectiveness of GP forecasting, we utilize $200$ samples for training and propose to forecast $\{500, 1000, 1500, 2000\}$ samples out of the training dataset. Also, we consider different SNRs in $\{50, 100, 150\}$. The forecast samples are integrated into PNN model and other benchmark models for comparisons. Fig. 6 demonstrates the results with respect to the forecast points. Due to the space limit, we only show results of GP-based PNN and GP-based Resnet for comparisons. The patterns of results for other benchmark methods are similar to GP-based Resent. We observe that GP-based PNN performs much better than GP-based Resnet, which is consistent to observations in the previous subsections. As the number of forecast data points increases, the prediction MSE value generally decreases for both PNN model and Resnet model. It implies that our GP forecasting produces accurate forecast points, suitable for all prediction models. When the number is larger than $500$, the results don't show a significant improvement. The reason is that forecast points far from the training set boundary are with low confidence, which don't have a big impact on learning results.

## 5 CONCLUSION

The lack of physical understanding makes it difficult for planning, monitoring, and control of IoE. To learn governing physical equations from data, the paper first presents a fundamental challenge for data-driven methods like symbolic regression and sparse regression. Data-driven only methods cannot guarantee to find the exact physics because many low-complexity expressions can perfectly fit the data. To resolve the issue, we propose a complete set of constraints based on physical principles to restrict search space. Thus, an efficient physical neural network (PNN-RISE) is developed. Finally, the FOCV supports the scenario when (partial) physical constraints are unknown. The forecast of data beyond the training dataset enhances model robustness with few extra costs. Results on physical system datasets show the superior performance of proposed methods.

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

# A    FIGURES

| Solutions | Case 1 | Case 2 | Case 3 | Case 4 |
|---|---|---|---|---|
| Physical Truth | $y = x_1x_2 + x_2x_3$ 
 $\mathcal{L} = 0.00$ | $y = 0.1x_1x_2 + 0.3x_2x_3$ 
 $\mathcal{L} = 0.00$ | $y = x_1x_2 + x_2^2x_3$ 
 $\mathcal{L} = 0.00$ | $y = x_1x_2 + x_2x_3 + x_1x_3$ 
 $\mathcal{L} = 0.00$ |
| Local Optimum 1 | $y = 0.0024x_1x_2 +$ 
 $2.0793x_1^{0.4945}x_2x_3^{0.4723}$ 
 $\mathcal{L} = 0.002$ | $y = -1.3192x_1^{0.6051}x_2^{0.3718}x_3^{0.3445}$ 
 $+1.1166x_1^{0.3569}x_2^{0.8011}x_3^{0.6605}$ 
 $\mathcal{L} = 0.003$ | $y = 6.2041x_1^{0.4331}x_2x_3^{0.9930}$ 
 $8.2098x_1^{0.4833}x_2x_3^{0.9993}$ 
 $\mathcal{L} = 0.021$ | $y = 1.8595x_1^{0.7805}x_2^{0.6445}x_3^{0.6502} +$ 
 $0.3481x_1^{0.4670}x_2^{-0.5860}x_3^{0.3862}$ 
 $+2.2448x_1^{0.1563}x_2^{0.7454}x_3^{0.7202}$ 
 $\mathcal{L} = 0.002$ |
| Local Optimum 2 | $y = 1.5205x_1^{0.4126}x_2^{0.9995}x_3^{0.3445}$ 
 $+0.9600x_1^{0.5369}x_2^{0.5920}x_3^{0.5693}$ 
 $\mathcal{L} = 0.001$ | $y = -2.6189x_1^{0.1352}x_2^{0.6277}x_3^{0.5121}$ 
 $+1.4484x_1^{0.2075}x_2^{0.8744}x_3^{0.6692}$ 
 $\mathcal{L} = 0.003$ | $y = 10.4416x_1^{0.7975}x_2x_3^{0.9997}$ 
 $-9.8486x_1x_2^{0.6719}x_3^{0.9996}$ 
 $\mathcal{L} = 0.003$ | $y = 1.8902x_1^{0.6901}x_2^{0.4758}x_3^{0.7413}$ 
 $1.0969x_1^{0.6724}x_2^{0.8458}x_3^{0.5806}$ 
 $1.0300x_1^{0.1140}x_2^{0.2962}x_3^{0.9306}$ 
 $\mathcal{L} = 0.002$ |
| Near Global Optimum | $y = x_1x_2 +$ 
 $x_2^{0.9900}x_3^{0.9900}$ 
 $\mathcal{L} = 0.026$ | $y = 0.0990x_1x_2 +$ 
 $0.3100x_2^{0.9900}x_3^{0.9900}$ 
 $\mathcal{L} = 0.004$ | $y = x_2^{1.9800}x_3 +$ 
 $x_1^{0.9900}x_2^{0.9900}x_3^{0.9900}$ 
 $\mathcal{L} = 0.056$ | $y = x_1x_2 + x_2x_3 +$ 
 $x_1^{0.9900}x_3^{0.9900}$ 
 $\mathcal{L} = 0.018$ |

Figure 7: Absolute mismatches between the recovered functions and the physical ground truth.

**Sparse Fitting May not be Physical Verifiable** For example, to learn the function $y = x_1x_2 + x_2x_3$ from data, different SR methods can easily get trapped in local optima. To see if these observation is special, we test different cases, where Fig. 7 shows a table summary of four different cases, with different coefficients. We also test different numbers for multiplication and summation. From all examples, one can see that the recovered objective has sparse coefficients and negligible error (loss objective $\mathcal{L}$) at non-desirable local optimums. Specifically, the neural network naturally approximates towards reducing the learning error. Therefore, it could ignore the consistency with physical governing functions and update in the wrong direction, leading to overfitting and the local optima solution. Differently, for the last row, the learning parameter of 0.990 (or 1.980) is close to ground truth 1 (or 2), which means "Near Global Optimum" that has 99% accuracy of parameter recovery. However, the prediction error (highlighted in red) is larger than a direct approximation of a neural network (highlighted in blue), where the parameters are far away from the ground truth.

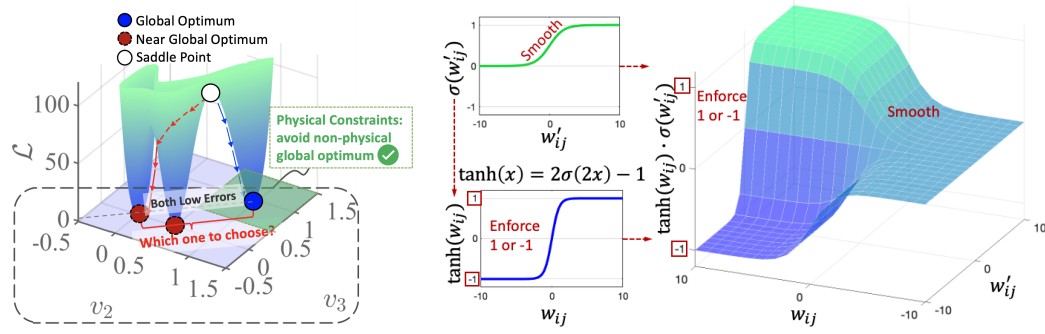

Figure 8: (Left:) Illustration of the possible saddle point, local optimum, and global optimum in physical equation learning. We observe that local optima can perform numerically as well as the global optimum (ground truth equation) since there is always a set of diversified equations to perfectly fit limited data. (Right:) Visualization of the arithmetic layer design function.

**Test Sparse Fitting for Power System Identification.** To make the realistic case study rather than toy examples, we use a power system case study and visualize the objective surface in Fig. 8 (left). Similarly, for a small parameter range, there are points achieving the global optimum or local optima that are very close to the global optimum. However, only one of them is the physical ground truth. For example, one can go from the red point to a non-physical solution with an error close to zero, even when the sparsity regularization is enforced. From the observation, we can see that "good" sparse fitting is insufficient to avoid overfitting in learning physical equations.

**Visualization of Neural Arithmetic Design in PNN.** We plot the surface of the neural arithmetic design in Fig. 8 (right). The visualization shows that the multiplication of tanh and sigmoid functions not only generates a smooth optimization surface but also naturally results in limited parameters.

**Illustration of RISE Constraints on PNN.** For illustrating the completeness of these principles, Fig. 9 presents the relationship among the four RISE concepts.

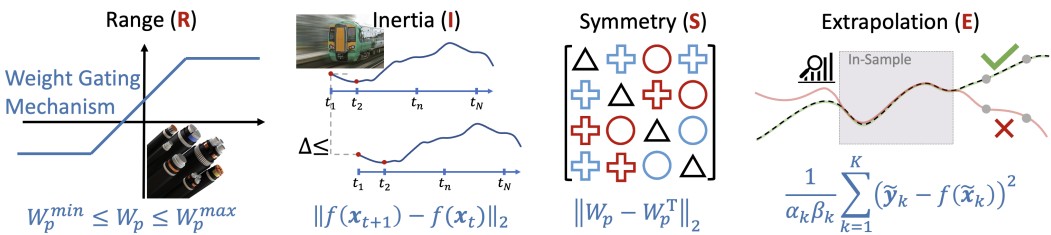

Figure 9: RISE principles for physical laws.

## B  PROOF OF THEOREM 1

*Proof.* First, we define the biconvexity as follows (Gorski et al., 2007).

**Definition 1** (Biconvexity). *A function $\phi : A \times B \to \mathbb{R}$ is called biconvex if $f(a, b)$ is convex in $a$ for fixed $b \in B$, and $f(a, b)$ is convex in $b$ for fixed $a \in A$.*

Then, we prove PNN is biconvex. Recall for the PNN function $\boldsymbol{y} = f(\boldsymbol{x})$, the $k^{th}$ element can be written as: $y_k = \sum_j m_{jk} z_j = \sum_j m_{jk} \exp\left(\sum_{i=1}^N p_{ij} \ln(x_i)\right)$. $\forall j, k$, if $m_{jk}$ is known, we need to prove the convexity of $f$ with respect to $p_{ij}, \forall 1 \leq i \leq N$.

Since the linear combination $\sum_{i=1}^N p_{ij} \ln(x_i)$ is convex with respect to $p_{ij}$, and the exponential function is convex, the convex function composition implies that $\exp\left(\sum_{i=1}^N p_{ij} \ln(x_i)\right)$ is convex with respect to $p_{ij}$. Therefore, $\forall j, k$, if $m_{jk}$ is known, the function $y_k = \sum_j m_{jk} \exp\left(\sum_{i=1}^N p_{ij} \ln(x_i)\right)$ is convex with respect to $p_{ij}, \forall 1 \leq i \leq N$. On the other hand, $\forall i, j$, if $p_{ij}$ is known, $y_k = \sum_j m_{jk} \exp\left(\sum_{i=1}^N p_{ij} \ln(x_i)\right)$ is a linear combination of $m_{jk}$. Therefore, $y_k$ is convex with respect to $m_{jk}$ given $p_{ij}$.

Therefore, PNN function $\boldsymbol{y} = f(\boldsymbol{x})$ is biconvex with respect to the logarithm layer and the linear summation layer. The biconvexity can lead to an efficient alternate convex search (ACS) (Gorski et al., 2007) to seek for optimal solutions. Namely, ACS will fix one part of parameters to search for the other part iteratively. Due to biconvexity, in each iteration, ACS will solve a convex problem. Then, Corollary 4.10 in (Gorski et al., 2007) guarantees that the searched points will converge to stationary points of $f(\boldsymbol{x})$. $\square$

## C  DETAILS OF DATASETS AND EXPERIMENT SETUPS

**Synthetic dataset:** generating random samples of $\boldsymbol{x}$ from Gaussian distribution $\mathcal{N}(0, 1)$, and obtain the outputs by $y = c_1 x_1 x_2 + c_2 x_1^2$.
**Power system datasets:** IEEE provides standard power system models, including the grid topology, parameters, and generation models, etc., for simulations. The model files and the simulation platform, MATPOWER (MATPOWER, 2020), are based on MATLAB. For simulations, the load files are needed as the inputs to the systems. Thus, we introduce power consumption data from PJM Interconnection LLC (PJM Interconnection LLC, 2018). Such load files contain hourly power consumption in 2017 for the PJM RTO regions. With the above data, MATPOWER produces the system states of voltages and nodal power injections. Then, we utilize the voltage as input and the nodal power as the output to learn the physical mapping, for which the true physical function is used for evaluations (Yu et al., 2017). To diversify the system types, we utilize the mass-damper system and aerodynamic system to further validate our results.
**Motion dynamics dataset: mass dampers of in-plane wind turbine.** The physical equation of the mass-damper system is representative of aircraft motion for analysis of system dynamics (van der Schaft, 2017; McRuer et al., 2014). Moreover, the active tuned mass dampers are implemented to control in-plane blade vibrations (Fitzgerald et al., 2013). Using MATLAB, we simulate the dynamic process of the mass-damper system with 10 nodes and obtain the measurements of velocity and momentum for the test of physical equation discovery.
**Aerodynamic dataset: aircraft weight estimation:** The aircraft sensor data is collected from the

aircraft simulation in MATLAB and the corresponding aerodynamic model is illustrated in (Chen et al., 2018). In this case, the goal is to recover the aerodynamic model of gross weight and angle of attack in a cambered airfoil.

For the implementation of machine learning methods, the prepared data samples are split as 60% for training the model, 15% for validation of model hyperparameters, and 25% for testing. Specifically, hyperparameters of the benchmark methods are selected through $k$-fold cross-validation. For SVR, the kernel is chosen from the candidates that include polynomial kernel ($2^{nd}$-degree and $3^{nd}$-degree), radial basis function (RBF) kernel, and hyperbolic tangent kernel. The box parameter is similarly chosen from $\{1 \times 10^{-3}, 5 \times 10^{-3}, 1 \times 10^{-2}, 5 \times 10^{-2}, 1 \times 10^{-1}, 5 \times 10^{-1}, 1 \times 10^{0}, 5 \times 10^{0}, 1 \times 10^{1}\}$.

## D  RESULTS OF THE SYNTHETIC DATASETS

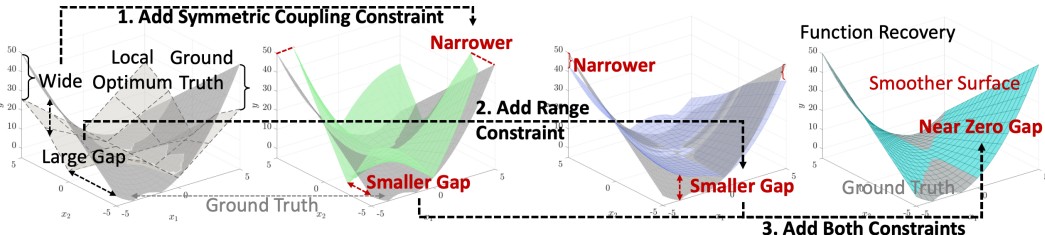

Figure 10: Synthetic dataset: visualization of the function recovery with RISE constraints.

In order to show the performance of using different physical constraints, we provide an ablation study of the proposed NN. We use the synthetic example to visualize the effect of constraints. The left-hand side of Fig. 10 compares the local optimum (transparent gray surface) learned by NN with the true function (solid gray surface). Though there are some overlaps, the large gap between the two surfaces causes large errors for generalization. By adding constraints, we observe in the middle of Fig. 10 that the variable coupling constraint rotates and stretches the surface to the correct shape like the maximum and minimum values of the surface. On the other hand, the parameter range constraint smoothes the surface of the local optimum. Together, the constraints push the recovered function (cyan surface) towards the ground truth, where the overlap in Fig. 10 (right) shows the physical exactness in not only feature components but also the corresponding parameters.

