# OpenReview forum: "WHAT TO DO IF SPARSE REPRESENTATION LEARNING FAILS UNEXPECTEDLY?"
_ICLR.cc/2022/Conference — ICLR 2022 Submitted_

### Official Review · Reviewer_7VKA · 2021-10-30

**Correctness:** 2
**Technical Novelty And Significance:** 2
**Empirical Novelty And Significance:** 2
**Recommendation:** 3
**Confidence:** 4

**Main Review:**

While the carried out work seems interesting, the paper does not allow to clearly highlight the contributions due to a partial presentation. The paper would have benefitted from a comprehensive presentation that provide the tangible elements to explain all the building blocks of this work. For instance, the so-called physical neural network (PNN) is presented in terms of the elements defining it. However, it is not clear its overall architecture, and most importantly the optimization method. While Proposition 1 give some elements, about the biconvexity and “the searching algorithm that can find the stationary points of PNN”, it is never explained what is the used algorithm. Moreover, in the description of the method (Section 3), we can find some incorrect information, such as “p_ij is the index number of x_i” (it is its power), “summation/abstraction”, … Finally, sentences like “After providing a flexible design to limit possibilities of local optima” are not founded with theoretical proofs.

The title is not appropriate. This paper is not about sparse representation learning, which is a more general topic than the one addressed in this work. It is on learning  physical equations with symbolic regression. Moreover, the authors never addressed the question in the title of this paper.

From the beginning of the paper, the authors emphasize on the main motivation of this work, which is Internet of Everything (IoE) that becomes IoT in page 2. However, almost nothing in the paper is related to this motivation, as all developments are not related to IoE, as well as most experiments.

The experiments are not convincing. The authors compare the proposed method to 4 other methods, including 2 methods that are not for symbolic regression (SVR and ResNet). The only used methods that are related to this work are SINDy (sparse identification of nonlinear dynamics) and equation learner. The authors need to provide a comprehensive experimental analysis, with a comparative analysis on several recent methods from the literature, such as:
- Raissi, M., Perdikaris, P., & Karniadakis, G. E. (2019). Physics-informed neural networks: A deep learning framework for solving forward and inverse problems involving nonlinear partial differential equations. Journal of Computational Physics, 378, 686-707.
- Lusch, B., Kutz, J. N., & Brunton, S. L. (2018). Deep learning for universal linear embeddings of nonlinear dynamics. Nature communications, 9(1), 1-10.
- Raissi, M. (2018). Deep hidden physics models: Deep learning of nonlinear partial differential equations. The Journal of Machine Learning Research, 19(1), 932-955.
- Udrescu, S. M., & Tegmark, M. (2020). AI Feynman: A physics-inspired method for symbolic regression. Science Advances, 6(16), eaay2631.
See also: Willard, J., Jia, X., Xu, S., Steinbach, M., & Kumar, V. (2020). Integrating physics-based modeling with machine learning: A survey. arXiv preprint arXiv:2003.04919, 1(1), 1-34.

The numbering in the body text does not correspond to the one in figure 1, making it difficult to understand.

There are some spelling and grammatical errors that can be easily identified and corrected, such as “we propose an new training “, “To see if these observation is special”, “bus actual knowledge”, “proof is inlcuded in Appendix”, “for each sub-regions”, “by combing their initials”, “leads an important extrapolation”, “the data in represented as”,  “our method is different as active learning”, “Visualization of the layer design equation ??”, “RISE Principals for Physical Laws”, “the load files are need as”


**Summary Of The Paper:**

This paper is about learning  physical equations with symbolic regression with neural networks.

**Summary Of The Review:**

We think that this paper is not of sufficient quality to be accepted in ICLR, for at least the reasons mentioned in the Main Review section.

---

### Official Review · Reviewer_8Azy · 2021-11-03

**Correctness:** 2
**Technical Novelty And Significance:** 2
**Empirical Novelty And Significance:** 2
**Recommendation:** 5
**Confidence:** 3

**Main Review:**

The proposed neural network is of very specific design resulting in the weighted sums of products of original inputs (and their integer powers). Sparsity in parameters is enforced using the tanh and sigmoid transformations of weights. Additionally, model parameters are constrained by domain knowledge induced physical principles like penalties on symmetry and smoothness and limits and inequalities of parameter values. Final piece of the framework is regularization term based on outside-of-the-sample predictions. Even though the individual tricks are likely used before, to the best of my knowledge this particular combination/instance of the neural network is unique so far. The empirical evaluation is sound and shows superior fit and physics recovery. Just wondering why is 0.990 (or its integer multiple like 1.980) "dominating" the Near Global Optimum row of the Figure 2.?  The article is quite descriptive and detailed. Just I think the ablation study would better serve as a part of the main article, and instead of the visual - some quantitative measure of contribution of each of the piece would be appreciated. Text is clear, with just few typos like "bus actual knowledge" or "different as active learning".

UPDATE
After insight into other reviewers' comments and authors' responses, I tend to agree with concerns on theoretical soundness of certain aspects and need for more appropriate experimental benchmarks. Therefore I am reducing the initial score.

**Summary Of The Paper:**

A framework for learning succinct and humanly interpretable descriptions of physical systems from data is outlined. The main focus of the approach is eliminating/avoiding local optima, which is achieved through both architectural design of the model and modification of learning algorithm. Empirical evaluations conducted on simulated data suggest improved error metrics as well as recovery of ground truth equations.

**Summary Of The Review:**

I think that study described in this paper would be beneficial for the conference readers, and especially the complex systems modeling community.

---

### Official Review · Reviewer_Lobz · 2021-11-08

**Correctness:** 4
**Technical Novelty And Significance:** 3
**Empirical Novelty And Significance:** 2
**Recommendation:** 6
**Confidence:** 4

**Main Review:**

### Strength
1. This paper shows a nice connection between machine learning models and a real-world application, i.e., estimating physical equations.
2. This paper has been well motivated. Because directly using a deep neural network, physical equations might not be learned well, due to a number of reasons, such as too many local solutions, as shown in Figure 2.
3. The experiments have been done pretty comprehensively, including those on synthetic data and those on real physical systems.

### Weakness
1. Some details of the experimental settings are missed. For example, in experimental section support vector regression (SVR) is used as baseline. What kernels are used in SVR? and other parameters, such as the box parameter?

**Summary Of The Paper:**

This paper studies the problem to learn physical equations by neural networks. To address the limitations of traditional methods using sparsity to learn physical equations, this paper proposes to use the physical neural network as a container, with additional constraints to explore “Range, Inertia, Symmetry, and Extrapolation (RISE)”. Using the proposed method, it is expected that less local minima will be explored for better solution. Experiments have been conducted to support the presented method.

**Summary Of The Review:**

This paper solves the problem of estimating physical equations by a physical neural networks (PNN), with the improvements to take into account “Range, Inertia, Symmetry, and Extrapolation (RISE)” that are important property of physical equation learning. The paper is well motivated and the method has been clearly described. Experimental results on both synthetic and real data support the presented method.

---

### Official Review · Reviewer_VZaq · 2021-11-09

**Correctness:** 2
**Technical Novelty And Significance:** 3
**Empirical Novelty And Significance:** 2
**Recommendation:** 3
**Confidence:** 3

**Main Review:**

### Strengths:
- The problem tackled is very important and indeed not studied as much as it should by the community.
- The experiments showing that existing approaches fail to recover the underlying equation structure for simple systems illustrates this well.
- The idea of relaxing the polynomial layer of PNN with tanh and sigmoid functions is very interesting, appealing and seems original to me.
- The idea of proposing a unifying formalism for physical constraints such as RISE is interesting and should be helpful.
- The proposed method is show-cased in a number of diverse experiments and compared to relevant SOTA methods.

### Weakness
In my opinion, despite these positive points, the article has major weaknesses in terms of theoretical and experimental soundness and clarity. Given the multiple clarity concerns, it is possible that I misunderstood some things, so please let me know it this is the case. Here follows my more detailed concerns.

**Concerning the proposed method**
1. It is not clear to me how you go from outputs in the  {-1, 1} range to the {-K, … K} range.
2. Furthermore, I must say that the Laurent polynomial structure of the method proposed does not seem expressive enough to model many important physical systems, which do not have a polynomial structure. This seems to me as a weakness compared to previous methods such as EQL and SINDy, which are built on top of an arbitrary dictionary of basis functions (trigonometric functions, exponentials, etc.). In this sense, the hypothesis space of PNN is included in SINDy and EQL ones.

**Concerning the RISE penalty**

3. Concerning the RISE constraint, while the formalism is interesting, its motivation lacks illustrative examples other than the power line example. Where else do we see the symmetry constraint and how are all these constraints implemented in the 4 experiments presented in the paper? How are constrained weights chosen? For example, I imagine you don’t symmetrize all weights, do you? Furthermore, the constraints presentation is not well connected to existing penalties from the statistical learning literature. For example, inertia constraint is usually modeled by smoothness inducing regularization (e.g. smoothing splines [1], dynamic system constraints like in system identification [2]).
4. Also, I was a bit surprised and confused because in section 3.1 (second paragraph) it is said that existing methods such as SINDy add additional hyperparameters (LASSO penalty weight) and that the proposed method does not have this flaw. But the RISE penalty also has a penalty hyperparameter to be tuned.

**Concerning the theoretical results**

5. It is said just before proposition 1, on page 5, that "such a design converts the non-convex multiplication to a convex form of linear summation for NN training". I imagine that this refers to the exp-sum-log rewriting of the polynomial $z_j=\prod x_i^{p_{ij}}$ at the end of page 4 as $\exp(\sum p_{ij} \ln x_i)$. Am I correct? If this is the case, I cannot agree with the statement as these two expressions are exactly equal and hence, both convex in $p_{ij}$ (they are the same function). There seems to be a confusion here because $z_j=\prod x_i^{p_{ij}}$ is non-convex in $x_i$. Please, let me know if I missed something here.
6. Concerning proposition 1, the "searching algorithm" is not defined (one has to go to the appendix to understand that you mean alternate convex search).
7. More importantly, I cannot agree with the results or at least not with its relevance here. Indeed, in addition to the linear weights $m_{jk}$, the optimization variables of the proposed PNN model are not the $p_{ij}$’s, but rather the auxiliary $w_{ij}$ and $w’_{ij}$ weights, and it happens that the problem is not biconvex wrt them (as evidenced clearly by figure 7). I do agree however that the problem is biconvex wrt $p_{ij}$’s and $m_{jk}$ and that ACS would work if one would optimize these variables instead. However this is not what the paper proposes.
8. At the end of the proof of theorem 1 (page 7) you state that "the piecewise strong convexity from RISE regularization guarantees that we can find the global optimal point for each sub-regions, and the global optima satisfies constraints for physical parameters". While you proved indeed the strong convexity and while this implies that the global minimum can be achieved with gradient descent, the optimization problem solved has been relaxed and, as such, has no hard constraint imposed, but rather a penalty term. Hence, nothing guarantees that the problem solution respects the constraints (not true for low values of $\lambda$ for example).
9. Also inside the proof of theorem 1 you replace the L0 norm in the symmetry constraint by a L1 norm, but this is not said in the theorem statement. For clarity (and rigor), you should define the symmetry constraint directly in terms of L1 norm, saying that it is relaxing an ideal L0 norm which is not used in practice. The proof is not the right place to do the switch.

**Concerning the experiments**

10. As evidenced by sentences like: "Subsequently, can we find the global optima and can it represent the true physics?" the paper seems to imply that physical solutions correspond to global minimum (of the training loss?) and that local minima are bad solutions. This is however not really shown in experiments. Actually, figure 2 seem to indicate the opposite: physical models have higher loss.
11. Overall, the experimental settings and protocols of all results are not sufficiently explained, which makes it impossible to interpret and assess them. For example, in figure 2, what are the hypothesis spaces considered? What method is used to obtain those scores (SINDy, EQL, …)? How are hyperparaameters selected (specially important as this is highlighted as a key pain that the proposed method should alleviate)? Same goes for figure 7 in appendix A.
12. Also, while you say at the end of page 3 that $\mathcal{L}$ is the loss, it is unclear in figure 2 whether the reported $\mathcal{L}$ is the training loss, the test loss, the cross-validated loss, … It is also never stated how many data points are used and how the data is split.
13. Experimental details are also highly insufficient in figure 4: how is the data split (could not find even in appendix)? Do the scores correspond to the training or testing error? How is optimization carried in Resnet, PNN-RISE (optimizer, number of iterations, batch size, learning rate, definitions of convergence, etc.) What dictionary of basis functions is used for SINDy and EQL? Does it include the true structure? And yours?  How are hyperparameters tuned? These details are very important to assess the fairness of the comparison here. It is indeed difficult to understand what in the proposed method allows it to beat the others given that its hypothesis space is theoretically included in the others. Maybe the RISE constraint or the training procedure? The experiments don’t really help to answer this question.
14. Also, what is the difference between PS1/PS2 (not explained in the appendix).
15. How many runs are used to compute the recovery rates?
16. You say that "EQL÷ easily fails in large power system case" but you never state the dimension of the problems considered.
17. In figure 5, a "limited data" case is considered, but it is not explained what this means exactly? How much less data is considered and how much did we have in the beginning? Also, what is the "out-of-scale invariants" case?

**Other clarity concerns**

18. It is confusing to say in the last paragraph of page 3 that "with integer coefficients or not" while the PNN model with integer coefficients was not introduced yet
19. Likewise, the proof of theorem 1 seems to be the first time the reader learns that the loss considered in the paper is always the MSE.
20. What is meant by "we complete the function y = f(x)" on top of page 5?


[1] Green, P. J.; Silverman, B.W. (1994). Nonparametric Regression and Generalized Linear Models: A roughness penalty approach. Chapman and Hall.

[2] L. Ljung. System Identification: Theory for the User. Prentice-hall, 1987.

**Summary Of The Paper:**

This paper tackles the problem of model inference and prediction, while preserving physical correctness. This is an interesting and important problem in supervised learning, for interpratability and generalization reasons. It is also quite important in model-based reinforcement learning, in the context of nonlinear system identification.

The authors start by challenging existing sparse regression approaches (such as SINDy) and symbolic regression approaches (such as EQL):
- Because they lead to non-physical solutions with low loss,
- Because they depend on hyperparamters

Hence they propose what is coined a Physical Neural Network (PNN), where:
-  layers have a Laurent polynomial shape with learnable coefficients and powers,
- weights corresponding to the layer input powers are decomposed into a tanh activation multiplied by a sigmoid activation.

This last design choice allows to enforce some kind of sparsity (when sigmoid converges to 0) and constrains powers in the range {-1, 1} when active (because of tanh). Furthermore, it ensures differentiability of the integer valued layer-input powers.

Additionally, the authors also propose to penalize the model training with what they call a RISE constraint. The latter is meant to enforce physically plausible range, inertia, symmetry and extrapolation properties in the inferred model.

**Summary Of The Review:**

Although the paper highlights weaknesses of existing approaches for a very important learning problem and proposes interesting novel modeling ideas, it has major weaknesses in terms of theoretical and experimental soundness and clarity. Hence, I cannot recommend it for publication at its current state.

---

### Official Review · Reviewer_XyFH · 2021-11-15

**Correctness:** 2
**Technical Novelty And Significance:** 3
**Empirical Novelty And Significance:** 2
**Recommendation:** 3
**Confidence:** 3

**Main Review:**

## Strengths

- Model parsimony (often synonymous with sparsity but not necessarily) has been an overarching goal in the physical sciences. Ensuring that the model is interpretable and physically consistent is also crucially important. While this has been widely recognized in the system identification community, it is my impression that this problem is not studied as much as it should in the machine learning community. The contribution by the present authors goes in this direction.
- The design of the polynomial layer in PNN, combining sigmoïd functions for sparsity and hyperbolic tangent for integer exponents is interesting and, to the best of my knowledge, is original.
- Although I don't find it flexible enough, the RISE framework is a step in the good direction to enforce realizability conditions in the model identification procedure.

## Weakness

Despite these positive points, this contribution suffers from certain limitations. The most important ones are discussed below.

### Concerning the proposed method
- The architecture of the network implicitly assumes that the unknown function $\mathbf{f}$ is a polynomial. What if it is not (e.g. the nonlinear pendulum or the Kuramoto model for oscillator synchronization)? Does it identify the Taylor (or Laurent or Padé) series expansion of $\mathbf{f}$?
- What if the system is only partially observed? Does the identified model include closure terms?
- Are $c_1$ and $c_2$ in the sigmoïd and hyperbolic tangent functions free parameters or are they actually hyperparameters?

### Concerning the RISE penalty

Despite what is stated in the introduction, PNNs are not restricted to systematically obeying the physical properties and actually have hyperparameters that need to be tuned! The constraints are weakly imposed by penalizing the cost function. Hence, the coefficient for this penalization is a hyperparameter and, if the latter is too small, then the identified model will not necessarily satisfy the constraints.

Moreover, this framework is not as universal as what is claimed by the authors, in particular, the Symmetry part:
- Symmetry in phase space does not necessarily imply symmetric coefficients in the equations. This is easily exemplified by the harmonic oscillator $\dot{x} = y$ and $\dot{y} = - x$ where the coefficient are anti-symmetric despite the trajectory in phase space being a circle.
- Rather than focusing on symmetric coefficients, a better approach would be to enforce equivariance, i.e. dynamics of the form $\mathbf{S} \dot{x} = f(\mathbf{S} x)$.
- Although I cannot remember of any proof, I believe that symmetry in the coefficient actually implies some form of conservation, something which might be useful for the power systems considered in this contribution but is too restrictive in general.

### Concerning the theoretical results

Most of my concerns have already been raised by reviewer VZaq.

### Concerning the experiments
- The test cases used for the comparisons are non-standard (at least from my system identification point of view). Although it may have been done out of simplicity by the authors, using poorly documented benchmarks is a surprising choice when introducing a new method.
- Experimental details are insufficient. How much data is used? How is it split? How are Resnet and PNN trained (optimizer, number of epochs, batch size, learning rate, etc)? Which functions are included in the library for SINDy and EQL? Which algorithm/formulation of SINDy has been considered?
- While the method proposed by authors explicitly takes into account some constraints, only the vanilla version of SINDy has been considered (to the best of my understanding). Yet, as shown in [1, 2], SINDy can be extended to incorporate physical constraints in the model identification problem. Using SINDy with constraints would thus be a somewhat more fair comparison.
- System identification techniques are often used in the low-data limit (either because simulating the high-dimensional system for a sufficiently long period of time would require millions of computing hours or because data is gathered from extremely large experimental facilities that cannot be operated for long). How does the method behave in this low-data limit (e.g. when you have fewer data points that free parameters in the neural network)?
- Numerous systems have rapidly attracting manifolds in phase space which express themselves as nonlinear correlations in the time series. Consider the following model of self-sustained oscillators
$$\dot{x} = \sigma x - y - xz$$
$$\dot{y} = x + \sigma y - yz$$
$$\dot{z} = -z + x^2 + y^2$$
with $\sigma > 0$. It is easy to show that, very rapidly, $z(t) \simeq x^2 + y^2$, i.e. despite the negative term in the third equation, $z(t)$ actually grows over time because of nonlinear coupling. Given only data, simple techniques would identify the following equation for $z(t)$
$$\dot{z} = \alpha z - \beta z^3,$$
which is obviously wrong (although consistent with the data). How would your technique behave in such a situation where nonlinear cross-talk between variables is what drive the dynamics at first order?

### Miscellaneous
-  The quality of the English should not be taken into account when assessing the scientific quality of a contribution. Yet, there are a lot of typos in the paper that could have been avoided using simple spellcheck.
- It is unclear until relatively late in the paper that the loss function considered is the mean squared error. Although I do understand that a different metric could be used during the training stage, I think that making this clear early on is needed.

### References
[1] Loiseau & Brunton. Constrained sparse Galerkin regression. Journal of Fluid Mechanics, 2018

[2] Kaptanoglu et al. Promoting global stability in data-driven models of quadratic nonlinear dynamics. Physical Review Fluid, 2021.

**Summary Of The Paper:**

Robust methods for equation discovery respecting physics are of utmost importance for the application of data-driven methods in science en engineering. The present contribution proposes a new framework based on Physical Neural Networks (PNN) :
- it is assumed that the function $\mathbf{y} = \mathbf{f}(\mathbf{x})$ is polynomial (with possibly negative exponents),
- the network has only one hidden layer (as far as I understood),
- this hidden layer takes as input the state variables $x_i$ of the system and outputs $z_j = \exp \left( \sum p_{ij} \textrm{ln}(x_i) \right)$,
- In the above expression, $p_{ij} = \textrm{tanh}(c_1 w_{ij}) \cdot \sigma(c_2 w^{\prime}_{ij})$ with $\mathbf{W}$ and $\mathbf{W}^{\prime}$ parameters to be learned.
- The final output layer is a simple weighted sum, i.e. $y_k = \sum_j m_{jk} z_j$.

Sparsity is of the model results from the use of the sigmoïd activation function in the hidden layer while the hyperbolic tangent makes sure $p_{ij}$ is an integer (either positive or negative).

Along with this simple architecture, the authors propose the so-called RISE framework to include realizability conditions (e.g. physical constraints) the model needs to satisfy. These constraints are enforced only weakly by penalizing the loss function to be minimized in the training state. The overall model is tested on a number of non-standard benchmarks to illustrate its capabilities.

**Summary Of The Review:**

Robust methods for equation discovery respecting physics are of utmost importance for the application of data-driven methods in science en engineering. Although the method presented in this contribution seems interesting, its presentation lacks clarity. As discussed in the core of my review, comparisons with already existing techniques are biased (e.g. SINDy with constraints is not considered) and are not presented for representative cases or well-tested benchmarks. Other issues are also discussed in the review. Hence, I cannot recommend this work for publication.

---

### Decision · Program_Chairs · 2022-01-20

**Decision:**

Reject

**Comment:**

The topic and ambition of this paper has been judged as important by all reviewers. Yet there is
a consensus that the theoretical and experimental contribution is not strong enough to effectively
argue for an important novel lead which would justify publication at ICLR. For these rejections,
this paper cannot be endorsed for publication at ICLR 2022.